# Purification, Composition, and Anti-Inflammatory Activity of Polyphenols from Sweet Potato Stems and Leaves

**DOI:** 10.3390/foods14162903

**Published:** 2025-08-21

**Authors:** Huanhuan Zhang, Ling Zhang, Feihu Gao, Shixiong Yang, Qian Deng, Kaixin Shi, Sheng Li

**Affiliations:** 1Institute of Agricultural Products Processing, Chongqing Academy of Agricultural Sciences, Chongqing 401329, China; jdteazl@163.com (L.Z.); gfh740403@163.com (F.G.); yangsx1019@163.com (S.Y.); dqsean@163.com (Q.D.); skx@webmail.hzau.edu.cn (K.S.); 2Chongqing Academy of Chinese Materia Medica, Chongqing University of Chinese Medicine, Chongqing 400065, China

**Keywords:** sweet potato stems and leaves, polyphenols, macroporous resin, UHPLC-QE-MS/MS, anti-inflammatory, NF-κB, MAPK

## Abstract

Sweet potato stems and leaves (SPSL) are rich in bioactive polyphenols, yet their utilization remains underexplored. This study established an efficient method for SPSL polyphenol enrichment using macroporous resins, with UHPLC-QE-MS/MS characterization of the purified polyphenols (PP) and subsequent evaluation of anti-inflammatory activity. The results showed that NKA-II resin demonstrated the best purification effect on SPSL polyphenols among the six tested resins. The optimal enrichment procedure of NKA-II resin was as follows: loading sample pH 3.0, 4.48 mg CAE/mL concentration, and 80% ethanol (*v*/*v*) eluent. A total of 19 major compounds were characterized in PP, including 12 phenolic acids and seven flavonoids, with a polyphenol purity of 75.70%. PP pretreatment (100 and 500 μg/mL) significantly inhibited LPS-induced release of NO (by 40.62% and 68.61%), IL-1β (by 40.07% and 68.34%), IL-6 (by 40.63% and 52.41%), and TNF-α (by 52.29% and 73.76%) compared to the LPS group (*p* < 0.05), demonstrating potent anti-inflammatory effects. Western blot analysis revealed that PP exerted anti-inflammatory effects by inhibiting the NF-κB (via suppression of IκBα phosphorylation/degradation and blockade of p65 nuclear translocation) and MAPK (via inhibition of p38, ERK, and JNK phosphorylation) signaling pathways. These findings support the utilization of this agricultural by-product in functional food development, particularly as a source of natural anti-inflammatory compounds for dietary supplements or fortified beverages.

## 1. Introduction

Sweet potato (*Ipomoea batatas* L.) is a globally significant crop, with global production reaching 93.52 million tons in 2023 [1]. Sweet potato stems and leaves (SPSL), the above-ground parts of the sweet potato, exhibit yields comparable to their tuberous roots yet remain severely underutilized. Currently, only a small fraction of tender SPSL is utilized as vegetables, whereas over 95% of mature SPSL is systematically excluded from value chains as agricultural waste [2,3], leading to significant resource waste. Therefore, it is necessary to explore high-value utilization ways for SPSL to support the sustainable development of the sweet potato processing industry. In recent years, researchers have increasingly explored high-value applications for sweet potato by-products (leaves, stems, and peels). For example, Machado et al. [4] examined the impact of sweet potato leaves on dermatological health, highlighting their potential for dermatological uses. Carvalho et al. [5] investigated the use of sweet potato residues (leaves, stems, and peels) in biorefineries via torrefaction and pyrolysis. Zhang et al. [6] developed polyphenol-based edible films from peels for fresh fruit preservation. Of particular interest is SPSL, which contains abundant bioactive polyphenols, ranging from 2.73 to 12.46 g/100 g DW [7,8]. SPSL polyphenols include phenolic acids (e.g., chlorogenic acids, caffeic acid) and flavonoids (e.g., rutin, jaceosidin) [9], which contribute to various health benefits. Studies have confirmed that SPSL polyphenols exhibit antioxidant [10], anticancer [11], hypolipidemic [12], and hypoglycemic effects [13], positioning them as promising functional ingredients. Despite this potential, the lack of efficient purification methods and comprehensive pharmacological evaluations [14] hinders the high-value application of SPSL in the agri-food industry.

Purification techniques are vital for the development of plant polyphenols. Currently, diverse polyphenol enrichment methods have been explored, including supercritical fluid chromatography [15], membrane separation [16], preparative high-performance liquid chromatography [17], and high-speed counter-current chromatography [18]. However, these methods suffer from drawbacks such as low extraction efficiency for polar polyphenols, prolonged extraction cycles, and reliance on high-cost instrumentation, making them unsuitable for industrial-scale applications or requiring substantial investment for industrialization. Nowadays, macroporous resin adsorption has gained widespread adoption in polyphenol isolation because of its economical operation, high efficiency, and repeated usability [19]. Fan et al. [20] effectively enriched rosmarinic acid from Salvia przewalskii Maxim. leaves via X-5 resin, achieving a 3.65-fold increase in content from 12.41% to 45.30% with an 80.17% recovery yield. Hou et al. [21] enriched polyphenols from Vernonia patula by employing NKA-II resin, and the level of total polyphenols improved dramatically following large-scale column chromatographic purification, reaching 50.33% with 83.45% recovery. Macroporous resin chromatographic technique has been proven to be a realizable method for the industrial enrichment of polyphenols from plants. Nevertheless, there are numerous macroporous resins with diverse polarity, pore diameters, and specific surface areas, which can selectively adsorb target ingredients from crude plant extracts via hydrogen bonding, electrostatic interactions, complexation, and size exclusion. Therefore, it is critical to screen the optimal resin type for enriching SPSL polyphenols.

Inflammation is a complex biological cascade initiated by pathogen infection or tissue damage, characterized by typical features including redness, swelling, heat, pain, and impairment of tissue structure and function [22]. This intricate pathological cascade involves coordinated control of key inflammatory cytokines (such as tumor necrosis factor [TNF]-α, interleukin [IL]-1β, and IL-6) and secondary mediators (e.g., nitric oxide [NO], prostaglandins). The overproduction of inflammatory mediators plays a pivotal role in the progression of numerous acute and chronic inflammatory disorders, such as septic shock [23], rheumatoid arthritis [24], inflammatory bowel disease [25], and cancer [26]. The generation of these inflammatory mediators is mainly controlled via crucial intracellular cascades, particularly nuclear factor κB (NF-κB) and mitogen-activated protein kinase (MAPK) pathways [27,28,29]. Growing evidence indicates that plant polyphenols exhibit anti-inflammatory effects [30,31,32]. While limited studies have reported the anti-inflammatory activity of SPSL polyphenols [33,34], the underlying molecular mechanisms, particularly their regulation of key inflammatory pathways, including NF-κB and MAPK signaling, have not yet been fully elucidated.

Therefore, this study aims to (1) establish an efficient method for SPSL polyphenol enrichment using macroporous resins, (2) identify the compound composition of purified polyphenols (PP) by ultra-high performance liquid chromatography-Q exactive mass spectrometry/mass spectrometry (UHPLC-QE-MS/MS), and (3) evaluate the anti-inflammatory activity of PP using a lipopolysaccharide (LPS)-stimulated RAW264.7 cell model and investigate its mechanism of action.

## 2. Materials and Methods

### 2.1. Materials and Reagents

SPSL were picked from the High-Tech Agricultural Park in Chongqing on 30 September 2024, and the tuberous roots from the same source were harvested on 10 October 2024. The materials listed below were purchased from Hecheng New Materials Technology Co., Ltd. (Zhengzhou, China): S-8, NKA-9, NKA-II, AB-8, HP20, and DA-201 macroporous resins; Yuanye Bio-technology Co., Ltd. (Shanghai, China): caffeoylquinic acids (CQA) standards; and Sigma-Aldrich (St. Louis, MO, USA): acetonitrile and formic acid. Wuhan Pricella Biotechnology Co., Ltd. (Wuhan, China): RAW264.7 murine macrophage cell line; Beyotime Biotechnology Co., Ltd. (Shanghai, China): CCK-8 assay kit, nuclear and cytoplasmic protein extraction kit; Nanjing Jiancheng Bioengineering Institute (Nanjing, China): nitric oxide (NO) assay kit; ELK Biotechnology (Wuhan, China): mouse TNF-α, IL-6/1β/10 ELISA kits, total RNA extraction reagent, EntiLink™ 1st strand cDNA synthesis SuperMix, and EnTurbo™ SYBR Green PCR SuperMix; and ASPEN Biotechnology Co., Ltd. (Wuhan, China): SDS-PAGE gel preparation kit, RIPA total protein lysis buffer, BCA protein concentration assay kit, ECL chemiluminescence detection kit, and primary and secondary antibody dilution buffer. Additional reagents, all analytical grade, were sourced from Macklin Biochemical Co., Ltd. (Shanghai, China).

### 2.2. Extraction of Polyphenols from SPSL

SPSL polyphenols were extracted as per the method described by Mu et al. [14] with minor modifications. Briefly, fresh SPSL was rinsed, vacuum freeze-dried, ground, and passed through a sieve of 40 mesh. A mixture of the SPSL powder and 80% ethanol (30 mL/g) underwent a 10 min ultrasonic-assisted extraction at 30 °C. The filtered solution was concentrated using a Heidolph Hei-VAP series rotary evaporator (Haidolph GmbH & Co. KG, Schwabach, Germany) at 45 °C, then centrifuged (8000 rpm, 15 min) to remove impurities from the aqueous phase. After equal-volume ethyl acetate extraction, the aqueous layer was recovered as crude polyphenol extract.

### 2.3. Total Phenolic Content (TPC) Determination

The TPC determination was performed by the Folin–Ciocalteu method [2]: Take 0.5 mL of the test solution, sequentially add 1 mL of 1 mol/L Folin–Ciocalteu reagent, 2 mL of 10% (*w*/*v*) Na_2_CO_3_ solution, and 1 mL of deionized water. Mix well, store in darkness at ambient temperature for 60 min, and measure the absorbance at 765 nm. A standard curve was plotted with chlorogenic acid standard solutions (0, 20, 40, 60, 80, and 100 μg/mL) versus absorbance, yielding the linear regression equation y = 0.0082 x − 0.0022 (R^2^ = 0.9999). The TPC was reported as mg/μg chlorogenic acid equivalent (CAE)/mL.

### 2.4. Purification of Polyphenols Extracted from SPSL

#### 2.4.1. Macroporous Resin Screening

The screening procedure was conducted as per the guidelines described by Mu et al. [14]. Six macroporous resins (S-8, NKA-9, NKA-II, AB-8, HP20, and DA-201) were comparatively assessed for polyphenol purification through the adsorption capacity (*Q_e_*, mg CAE/g) and desorption rate (*D*, %). The *Q_e_* and *D* were determined using Equations (1) and (2), respectively:

(1)$$Q_{e}\left(\mathrm{mg}\;\mathrm{CAE}/g\right)=\frac{\left(C_{0}-C_{e}\right)V_{0}}{m\times 1000}$$(2)$$D\left(\%\right)=\frac{C_{1}V_{1}}{\left(C_{0}-C_{e}\right)V_{0}}\times 100$$ where *C_0_* and *C_e_* (μg CAE/mL) denote the initial and equilibrium TPC of the sample solution; *V_0_* (mL) denotes the sample solution volume; *m* (g) denotes the resin mass; *C_1_* (μg CAE/mL) denotes the TPC of the desorption solution; and *V_1_* (mL) denotes the volume of eluate.

#### 2.4.2. Static Adsorption–Desorption Test

A total of 2.000 g of pretreated screened resin and 50 mL of sample solution were combined in a 150 mL Erlenmeyer flask. Static adsorption was performed via shaking at 25 °C and 130 rpm for 12 h. The TPC was monitored every 30 min during the first 3 h, followed by hourly measurements. After two consecutive hourly measurements showed no significant variation in TPC, the final TPC at 12 h was directly measured to calculate *Q_e_*. The adsorbed-equilibrium resin was filtered, followed by a 10 mL deionized water rinse, and then elution was conducted with 50 mL of 80% ethanol (*v*/*v*). Desorption kinetics were analyzed under identical conditions. To evaluate the kinetic behavior of adsorption, pseudo-first-order (PFO) (Equation (3)) and pseudo-second-order (PSO) models (Equation (4)) [35] were fitted with the static adsorption data:
(3)$$\ln\left(Q_{e}-Q_{t}\right)=-t+\ln Q_{e}$$
(4)$$\;\frac{t}{Q_{t}}=\frac{1}{k_{2}Q_{e}^{2}}+\frac{t}{Q_{e}}$$ where *k_1_* (h^−1^) denotes the rate constant of the PFO model, *k_2_* (g·mg^−1^·h^−1^) denotes the rate constant of the PSO model, *Q_e_* (mg CAE/g) denotes the equilibrium adsorption capacity, and *Q_t_* (mg CAE/g) denotes the adsorption capacity at time t (h).

#### 2.4.3. Effect of Process Parameters on Adsorption–Desorption Performance [2]

Precise pH values (3.0, 4.0, 5.0, 6.0, 7.0, and 8.0 ± 0.1) of the sample solutions were obtained through the controlled addition of 1.0 mol/L HCl or NaOH solutions as required. For each pH condition, 2.000 g of pretreated screened resins were applied to 50 mL of the sample solution and underwent static adsorption for 3 h to detect the optimal pH.

The impact of the primary sample concentration was examined by diluting the crude SPSL extract to 0.40, 0.79, 1.55, 2.98, 4.48, and 5.53 mg CAE/mL, with the pH uniformly adjusted to 3. Identical adsorption conditions were applied across all concentration levels to determine the optimal solution concentration.

The resin was rinsed with 1 BV of deionized water subsequent to reaching equilibrium. Then, each resin (2.000 g) was desorbed with 50 mL of ethanol (50%, 60%, 70%, 80%, 90%, and 100%, *v*/*v*), respectively. Static desorption was conducted for 1 h to evaluate the ethanol concentration-dependent desorption behavior, and the preferred ethanol eluent concentration was identified.

#### 2.4.4. Dynamic Adsorption–Desorption Test

The pretreated screened resin (23.72 g) was wet-packed into a glass column (20 × 200 mm, 40 mL bed volume [BV]), filling the column to approximately 70% of its height. To determine the maximum treatment capacity of NKA-II resin for SPSL crude extract under optimal process conditions, the diluted crude extract (4.48 mg GAE/mL, pH 3.0) was passed into the column at 2 BV/h [36] under room temperature. When the TPC level in the effluent became 10% of the initial sample concentration, the breakthrough point was determined, and the corresponding breakthrough volume was calculated. Following a wash with 1 BV of distilled water, the resin was eluted with 80% ethanol (*v*/*v*), and the eluent volume was measured. Finally, concentrating the resulting solution was conducted by rotary evaporation at 45 °C, and then lyophilization was conducted to obtain PP.

#### 2.4.5. Pilot-Scale Purification Test

Under the optimal purification conditions, a pilot-scale purification test was carried out in a 100 cm-long, 10 cm-inner diameter column containing pretreated screened resin (70% BV), scaled up proportionally from lab-scale column chromatography.

### 2.5. Qualitative Analysis of PP by UHPLC-QE-MS/MS

The UltiMate 3000 (Thermo Fisher Scientific, Germering, Germany), paired with a Q exactive mass spectrometer (Thermo Fisher Scientific, Bremen, Germany), was utilized to identify PP as per the method of Luo et al. [9]. The chromatographic column used was a Hypersil Gold C_18_ column (2.1 × 100 mm, 1.9 μm). The mobile phases (A: 0.05% formic acid in water; B: 0.05% formic acid in acetonitrile) were introduced at 0.3 mL/min with 1 μL injections. The phase B gradient profile was optimized as follows: 5% (0 min), 60% (20 min), 5% (20.1 min), and 5% (30 min). A constant column temperature of 30 °C was maintained. Mass spectrometry parameters: HESI source; sheath gas flow rate: 40 a.u.; capillary temperature: 320 °C; aux gas heater temperature: 350 °C; aux gas flow rate: 10 a.u.; full MS/dd-MS^2^ with negative ion mode; full MS scan: *m*/*z* 100–1000, resolution 70,000; dd-MS^2^: resolution 17,500, fragmentation mode HCD, isolation window 4.0 *m*/*z*, normalized collision energy 35%, and mass tolerance: 5 ppm. MS^2^ fragment peaks were identified by comparison with publicly accessible data, known standards, and online databases.

### 2.6. Quantitative Analysis of Chlorogenic Acids in PP by HPLC

The Waters 1525 HPLC system (Waters, Milford, OH, USA), supplemented with a Zorbax Eclipse XDB-C_18_ column (4.6 × 250 mm, 5 μm) and a 2489 UV-Vis detector, was utilized for the quantitative analysis of chlorogenic acids. The analytical conditions and instrument parameters were set according to Chinese National Standard GB/T 43733-2024 [37]. Briefly, the flow rate of mobile phases (phase A, acetonitrile; phase B, 0.1% formic acid in ultrapure water) was at 1 mL/min with 10 μL of injection volume, while phase A changed as follows: 0 min (10%), 5 min (10%), 18 min (40%), 18.5 min (80%), 20 min (80%), 21 min (10%) and 30 min (10%). The column oven temperature was 30 °C, and the constituents were detected at 327 nm. Standard calibration curves were generated to measure the levels of individual components.

### 2.7. Anti-Inflammatory Activity Assessment

#### 2.7.1. Cell Culture and Sample Treatment

The RAW264.7 cells were cultivated in high-glucose DMEM with 10% FBS and 1% PS under standard circumstances (37 °C and 5% CO_2_). Accurately weigh 100 mg of PP and dissolve it in 10 mL of sterilized ultrapure water. The PP solution filtration was conducted through a 0.22 μm sterile filter membrane to prepare a 10 mg/mL master stock. Then, dilute the master stock with high-glucose DMEM to formulate target solutions with doses of 1, 10, 100, 500, and 1000 μg/mL.

#### 2.7.2. Cell Viability

The CCK-8 assay was utilized to estimate cell viability as per Qin et al. [38] with some modifications. Logarithmic-phase cell pellets were resuspended and then seeded in 96-well plates. After incubation for one night, the medium was removed, and 100 μL of several doses of PP (1, 10, 100, 500, and 1000 μg/mL) was applied. The same volume of high-glucose DMEM was introduced to the negative control group. Following incubation for an hour, CCK-8 (10 μL) was applied, and a 1 h incubation was conducted. The blank well contained an equal amount of high-glucose DMEM and CCK-8 reagent, but no cells. A microplate reader was employed to assess the absorbance at 450 nm, with blank-adjusted values used for viability calculations. The cell viability was measured using Equation (5):
(5)$$\mathrm{Cell}\;\mathrm{viability}(\%)=\frac{{\mathrm{OD}}_{\mathrm{experimental}}-{\mathrm{OD}}_{\mathrm{blank}}}{{\mathrm{OD}}_{\mathrm{control}}-{\mathrm{OD}}_{\mathrm{blank}}}\times 100$$

#### 2.7.3. NO and Inflammatory Cytokines Determination

The experimental groups were set as follows: negative control group, LPS group, and PP-treated groups. In the PP-treated groups, cells were pretreated with different concentrations of PP (20, 100, and 500 μg/mL) for 1 h, followed by stimulation with LPS (1.0 μg/mL) and co-incubation for 24 h. The LPS group was treated with LPS (1 μg/mL) for 24 h. The negative control group was incubated with an equal volume of culture medium for 24 h. The RAW264.7 cells were introduced into 6-well plates and cultivated overnight. Following group-specific treatments, the supernatant was collected for the determination of NO and inflammatory cytokine levels. The NO level was measured via a commercial NO assay kit, and inflammatory cytokine levels were quantified via commercial ELISA kits following the manufacturer’s guidelines.

#### 2.7.4. RT-qPCR

Total RNA isolation from treated cells was conducted utilizing the TRIzol reagent as per the manufacturer’s guidelines. The RNA level and purity were detected via a NanoDrop Lite Spectrophotometer (Thermo Fisher Scientific, Wilmington, MA, USA). Genomic DNA was eliminated through treatment with DNase I, and first-strand cDNA was generated via the EntiLink™ 1st strand cDNA synthesis kit. EnTurbo™ SYBR Green PCR SuperMix was utilized to conduct quantitative PCR on the QuantStudio 6 Flex Real-Time PCR System (Life Technologies, Carlsbad, CA, USA). The 2^−ΔΔCt^ method was used to assess target gene expression levels, with GAPDH acting as the internal reference gene. The primer sequences are listed in Appendix A.

#### 2.7.5. Western Blotting

Western blot analysis was carried out as shown by Shan et al. [39] with some adjustments. The protein levels were assessed by the BCA assay kit, with loading volumes adjusted accordingly. Proteins were then electroblotted onto 0.45 μm PVDF membranes after separation by SDS-PAGE gels. Following a 1 h blockade with 5% BSA, an overnight incubation of the membranes was conducted at 4 °C with specific primary antibodies. Following three wash cycles using TBST, an incubation of the membranes was conducted with the appropriate secondary antibody for 30 min. After utilizing TBST for three more washes, the ECL kit was utilized to observe the proteins. The images were captured using a 9000 F Mark Ⅱ scanner (Canon, Tokyo, Japan), and the bands were analyzed with AlphaEaseFC^TM^ 4.0.0.

### 2.8. Statistical Analysis

For every experiment, at least three repeated runs were performed, and the results were reported as mean ± SD. The cell culture data were statistically analyzed with GraphPad Prism 8.0 using paired *t*-tests for two-group comparisons and one-way ANOVA with Dunnett’s post hoc test for multi-group analyses. The statistical analysis of other data was performed using SPSS 27.0, with one-way ANOVA and Tukey’s post hoc test for multiple comparisons. All data were first tested for normality, and homogeneity of variance prior to ANOVA. *p* < 0.05 was deemed significant. Figures were plotted using Origin 2024, GraphPad Prism 8.0, and PowerPoint.

## 3. Results and Discussion

### 3.1. Macroporous Resin Screening

To select the optimal macroporous resin for purifying SPSL polyphenols, six resins with different polarities were systematically evaluated regarding their adsorption and desorption performance. Table 1 shows that the adsorption capacities of SPSL polyphenols by the tested resins are ranked as follows: NKA-II > S-8 ≈ AB-8 > HP20 ≈ NKA-9 ≈ DA201, with NKA-II resin exhibiting the maximum adsorption capacity. AB-8 macroporous resin exhibited an adsorption capacity of 22.45 ± 0.33 mg CAE/g, which closely matched the value (22.38 mg CAE/g) reported by Xi et al. [2]. Crucially, the NKA-II resin demonstrated significantly superior adsorption performance for SPSL polyphenols compared to AB-8 resin. In aqueous solutions, SPSL polyphenols display polar properties due to their abundance of phenolic hydroxyl moieties. According to the polarity matching principle of macroporous resins [20], polar resins exhibit stronger adsorption capacity for SPSL polyphenols, which may explain the superior adsorption performance of NKA-II resin. However, although NKA-9 and DA-201 are polar resins as well, their adsorption capacities for SPSL polyphenols remain relatively low. This observation indicates that other physical properties (pore size and specific surface area) have a substantial impact on the adsorption capability of macroporous resin, in addition to resin polarity [35]. According to Table 1, the NKA-9 and NKA-II resins have similar average pore sizes, but NKA-9 has a much smaller specific surface area than NKA-II. This may explain why the adsorption capacity of NKA-9 is significantly lower than that of NKA-II, as a larger specific surface area allows more polyphenol molecules to bind, resulting in stronger adsorption.

The desorption rate refers to the proportion of SPSL polyphenols eluted from the resin using a desorption solvent. As illustrated in Table 1, NKA-II resin demonstrates the highest desorption rate (90.50 ± 0.28%) among the six macroporous resins tested. This performance advantage likely results from the comparatively weak affinity of NKA-II resin for SPSL polyphenols [20], which enhances its regeneration efficiency. Among the six types of macroporous resins compared, NKA-II demonstrates optimal adsorption and desorption performance for SPSL polyphenols. Therefore, NKA-II resin was selected for further investigations.

### 3.2. Static Adsorption and Desorption Kinetics

The adsorption curve (Figure 1A) illustrates that NKA-II resin demonstrated rapid adsorption characteristics towards SPSL polyphenols, reaching the maximum adsorption rate within the initial 0.5 h while exhibiting a sharp increase in adsorption capacity. Over time, the adsorption rate followed a decreasing trend, while the adsorption capacity increased slowly, reaching equilibrium at 3 h with a maximum adsorption capacity of 29.20 ± 0.38 mg CAE/g. Prolonging the adsorption time had no significant impact on the capacity (*p* > 0.05). Thus, the optimal static adsorption time for NKA-II resin was determined as 3 h. This result aligned with the 3 h equilibrium time for AB-8 resin adsorbing sweet potato leaf polyphenols reported by Xi et al. [2]. The desorption curve (Figure 1A) reveals that polyphenols adsorbed onto NKA-II were rapidly released by ethanol eluent within 0.5 h, with a steep rise in desorption ratio. The desorption rate stabilized after 1 h, achieving an equilibrium desorption ratio of 90.54 ± 0.45%. Consequently, the static desorption time was set at 1 h.

To depict the adsorption more comprehensively and elaborate on the adsorption mechanism, the PFO and PSO models were utilized. Figure 1B,C and Table 2 exhibit the fitting outcomes, respectively. The adjusted R^2^ of the PSO kinetic equation (0.9971) was greater than that of the PFO kinetic (0.9671). Furthermore, the theoretical adsorption capacity of the PSO kinetic model (Qₑ = 30.175 mg CAE/g) was significantly closer to the experimental value (Qₑ = 29.2 mg CAE/g) than the PFO kinetic model (Qₑ = 18.654 mg CAE/g). These findings indicate that the PSO model was more appropriate for depicting the adsorption of SPSL polyphenols onto NKA-II resin, implying that the process is controlled by chemical adsorption [40]. This is because the adsorption behavior is governed by the functional groups on the NKA-II resin surface, including phenolic hydroxyl (-OH), ester (-COOR), and phenyl ring structures. The phenolic hydroxyl (-OH) drives chemisorption via hydrogen bonding with polyphenol polar groups (-OH, -COOH), while ester (-COOR) and phenyl rings mediate physisorption through hydrophobic interactions and π-π stacking. This finding agrees with that of Liu et al. [41], who detected analogous PSO kinetic behavior for chlorogenic acid adsorption on NKA-II resin. Additionally, Hou et al. [42] discovered that the adsorption process of phenolics derived from Danshen leaves onto NKA-2 resin also fitted well with the PSO model.

### 3.3. Effect of Process Parameters on Adsorption–Desorption Performance

#### 3.3.1. Effect of Sample Solution pH on Adsorption Capacity

Figure 2A illustrates that the polyphenol adsorption capacity of NKA-II resin exhibited a strong pH dependence. The maximum adsorption capacity (33.18 ± 0.72 mg CAE/g) was achieved at pH 3.0, followed by a gradual decline with increasing pH. This pH-dependent behavior originates from the acid–base equilibrium of phenolic compounds: under acidic conditions, polyphenols predominantly exist as non-dissociated molecules with appropriate polarity, enabling effective adsorption through hydrogen bonding and hydrophobic interactions with the resin matrix. Conversely, alkaline conditions induce deprotonation of the phenolic hydroxyl groups, weakening hydrogen bonding and reducing affinity for the moderately polar resin surface [43]. Thus, the sample solution pH was optimized at 3.0 for subsequent adsorption studies.

#### 3.3.2. Effect of Sample Concentration on Adsorption Capacity

Below 4.48 mg CAE/mL, a positive concentration-dependent relationship was observed, where the resin’s adsorption capacity exhibited pronounced growth (Figure 2B). However, further increases in solution concentration resulted in a markedly reduced adsorption rate enhancement, eventually reaching an adsorption equilibrium plateau. This phenomenon can be attributed to monolayer-dominated adsorption at lower concentrations, where abundant available binding sites facilitated rapid adsorption. As saturation was approached, the depletion of accessible adsorption sites coupled with increased diffusion resistance collectively contributed to the observed adsorption deceleration. Consequently, the optimal solution concentration was identified to be 4.48 mg/mL based on the equilibrium between adsorption efficiency and operational economy.

#### 3.3.3. Effect of Eluent Concentration on Desorption Ratio

Figure 2C illustrates that the desorption ratio progressively elevated with ethanol concentration in the 50–80% range. The difference between 80% and 90% ethanol concentrations was statistically insignificant (*p* > 0.05), yielding desorption ratios of 91.98 ± 0.26% and 92.06 ± 0.20%, respectively. At ethanol concentrations exceeding 90%, however, the desorption ratio unexpectedly decreased. This trend reversal may be attributed to the polarity-dependent dissolution mechanisms. Specifically, lower ethanol concentrations (higher solvent polarity index) exhibit reduced polyphenol elution efficiency, while excessively high ethanol concentrations promote competitive dissolution of alcohol-soluble impurities, thereby hindering target polyphenol recovery [21]. Through systematic evaluation of desorption performance, environmental impact, and process economics, the optimal ethanol concentration was established at 80% for applications.

### 3.4. NKA-II Macroporous Resin Column Chromatography

#### 3.4.1. Dynamic Adsorption andDesorption 

To accurately determine the loading capacity of the macroporous resin column for the sample solution, the breakthrough point was identified through analysis of the dynamic adsorption curve [44]. The dynamic adsorption behavior of NKA-II resin was investigated by loading SPSL polyphenol crude extracts (4.48 mg CAE/mL) using a 2 BV/h flow condition. During the initial loading phase (<160 mL) of the dynamic adsorption (Figure 2D), the effluent total phenolic concentration exhibited a gradual increase but remained below 200 μg/mL, indicating minimal polyphenol leakage. Upon reaching 260 mL (6.5 BV) of loaded solution, the effluent total phenolic concentration sharply increased to 451.90 ± 7.28 μg/mL, exceeding 10% of the initial concentration and signifying adsorption equilibrium. Further increases in loading volume resulted in rapid polyphenol breakthrough due to the saturation of resin adsorption capacity. These results demonstrated that NKA-II resin column chromatography can effectively process 6.5 BV of crude extract before reaching dynamic equilibrium.

Dynamic desorption experiments using 80% (*v*/*v*) ethanol under 2 BV/h elution conditions demonstrated efficient elution performance. For the dynamic desorption (Figure 2D), the majority of adsorbed polyphenols were recovered within the initial 80 mL (2 BV) of eluent, with complete desorption equilibrium attained at 120 mL (3 BV). The observed sharp elution peak without tailing indicated optimal compatibility between the eluent polarity and resin–solute interaction mechanism, which effectively disrupted hydrogen bonding and van der Waals forces between polyphenols and the resin matrix. This elution profile confirmed the suitability of 80% ethanol (*v*/*v*) for effective recovery of polyphenols from NKA-II resin. To reduce costs, 2 BV of 80% ethanol was used for elution, achieving a polyphenol recovery rate of 85.30 ± 0.85%. After concentration by rotary evaporation and vacuum freeze-drying, the polyphenol purity of the resulting purified product was determined to be 75.70 ± 0.92%.

#### 3.4.2. Pilot-Scale Purification

Based on dynamic adsorption/desorption results, 6.5 BV of SPSL polyphenol crude extract (4.48 mg CAE/mL, pH 3.0) was loaded into the NKA-II resin column at a flow rate of 2 BV/h. Next, the sample-loaded column was rinsed with 1 BV of distilled water, followed by elution with 2 BV of 80% ethanol using a 2 BV/h flow condition. The eluate was collected with an 83.53 ± 1.21% polyphenol recovery rate, then subjected to concentration and lyophilization to obtain a purified product with 73.20 ± 1.30% polyphenol purity. Compared to lab trials, the pilot-scale process showed slightly lower recovery and purity but remained within acceptable limits. This is consistent with the findings of Yang et al. [35], who reported a slight downward trend in the scaled-up enrichment of polyphenols from camphor tree seed using HJ-18 resin, compared with that in the small-scale batch. Moreover, the market price of NKA-II resin is affordable, and its reusability further reduces the actual cost. Considering factors such as cost, scalability, and product quality retention, this purification process proves highly viable for industrial applications.

### 3.5. Qualitative Analysis of PP by UHPLC-QE-MS/MS

The polyphenol compounds in purified polyphenols (PP) were analyzed by UHPLC-QE-MS/MS under negative ionization mode. Their retention times (Rt), molecular formulas, and key fragment ions were cross-referenced against available standards, digital databases (e.g., MassBank, SciFinder), and previously reported research. Twelve phenolic acids and seven flavonoids were detected and characterized, with their identification details listed in Table 3 and the mass spectra of precursor and fragment ions provided in Appendix A.

The identifications of 5-caffeoylquinic acid (CQA, Compound 2), 3-CQA (Compound 7), 4-CQA (Compound 8), 3,4-diCQA (Compound 13), 3,5-diCQA (Compound 14), 4,5-diCQA (Compound 15), and 3,4,5-triCQA (Compound 17) were achieved via the comparison of Rt and precursor ion peaks with those of authentic standards. Compound 3 had a parent ion [M-H]^−^ at *m*/*z* 339.0719, generated a fragment ion with an *m*/*z* of 177.0186 [C_9_H_5_O_4_]^−^, and was initially identified as aesculin [45]. The precursor ion (*m*/*z* 179.0341) of Compound 5 produced a main fragment at *m*/*z* 135.0440 [M-H-CO_2_]^−^ and was determined to be caffeic acid [46]. Compound 6 comprised the precursor ion [M-H]^−^ at *m*/*z* 161.0235, along with a fragment at *m*/*z* 133.0284 [M-H-CO]^−^, and was identified as 7-Hydroxycoumarin [9]. Compound 9 exhibited a parent ion [M-H]^−^ at *m*/*z* 367.1025, along with fragments at *m*/*z* 134.0358 and 191.0553, which represented the feruloyl moiety and quinic acid moiety, respectively. This compound was tentatively determined to be feruloylquinic acid [47]. Compound 12 showed a [M-H]^−^ peak at *m*/*z* 207.0657 and a fragment at *m*/*z* 163.0757 [M-H-CO_2_]^−^, identifying it as ethyl caffeate.

Compound 1 was characterized as quercetin based on the parent ion at *m*/*z* 301.0351 and the characteristic fragment ion at *m*/*z* 257.4846 [M-H-CO_2_]^−^ [9]. Compound 4 comprised the parent ion [M-H]^−^ at *m*/*z* 285.0437, along with a product ion at *m*/*z* 152.0106, and was presumed to be Kaempferol. Compound 10 had the parent ion [M-H]^−^ at *m*/*z* 609.1448 with a fragment at *m*/*z* 300.0287 [C_15_H_8_O_7_]^−^, and was identified as rutin. Hyperoside (Compound 11, [M-H]^−^ *m*/*z* 463.0872) generated a diagnostic MS/MS ion at *m*/*z* 300.0276 [M-H-C_6_H_10_O_5_-H]^−^, which matched the galactose neutral loss [48]. Compound 16, with a parent ion *m*/*z* value of 299.0555, generated an MS/MS ion at *m*/*z* 284.0320 [M-H-CH_3_]^−^, and was determined to be diosmetin. Compound 18 ([M-H]^−^ *m*/*z* 329.0662) produced characteristic fragments at *m*/*z* 314.0436 [M-H-CH_3_]^−^ and 299.0203 [M-H-CH_3_-CH_3_]^−^ and was identified as Jaceosidin. Compound 19 ([M-H]^−^ *m*/*z* 313.0714) generated an MS/MS ion at *m*/*z* 298.0487 [M-H-CH_3_]^−^ and was identified as pectolinarigenin [49]. Although the components of SPSL polyphenols differ among different cultivars, chlorogenic acids are universal in SPSL. Moreover, chlorogenic acids were found to account for a relatively high proportion of polyphenols in SPSL based on the total ion chromatogram and the literature reports [2,14]. Therefore, HPLC was subsequently employed for the quantitative analysis of chlorogenic acids in PP.

### 3.6. Quantitative Analysis of Chlorogenic Acids in PP by HPLC

It was confirmed that PP contained seven distinct chlorogenic acids (Appendix A), which were 4-CQA, 5-CQA, 3-CQA, 3,4-diCQA, 3,4,5-triCQA, 3,5-diCQA, and 4,5-diCQA. This compositional profile aligns with previously reported findings [2,7]. As listed in Table 4, the R^2^ values of all regression equations for the target compounds were greater than 0.99, showing significant linear concentration-peak area correlation (working ranges). Among the seven chlorogenic acids, 3-CQA showed the highest content (113.35 ± 1.26 mg/g), followed by 3,4-diCQA (75.52 ± 0.53 mg/g) and 3,5-diCQA (54.21 ± 0.28 mg/g). 5-CQA (17.39 ± 0.24 mg/g), 4-CQA (17.14 ± 0.34 mg/g), and 4,5-diCQA (11.37 ± 0.13 mg/g) displayed relatively lower levels, while 3,4,5-triCQA exhibited the lowest contents (9.03 ± 0.10 mg/g). However, Jung et al. [50] reported that 5-CQA and 3,5-diCQA are the major chlorogenic acids in SPSL and petioles. The discrepancies might be attributed to differences in the extraction solvent, the genotype of the tested cultivars, or the selective adsorption effects of macroporous resins. Additionally, Liu et al. [41] purified chlorogenic acid from *Ageratina adenophora* extracts using NKA-II resin, yielding a product with a chlorogenic acid purity of 22.17%, which is lower than the content of chlorogenic acids in this study.

### 3.7. Anti-Inflammatory Activity of PP

#### 3.7.1. Effect of PP on Cell Viability

To assess the immunomodulatory function of PP, the toxicity of PP to RAW264.7 cells was measured with the CCK-8 test. Figure 3A illustrates that no significant variation was found (*p* > 0.05) in cell viability between the control group (0 μg/mL) and PP-treated groups at 1–500 μg/mL. However, 1000 μg/mL PP markedly suppressed cellular proliferation (*p* < 0.0001). Consequently, the selection of three non-toxic doses (20, 100, and 500 μg/mL) was conducted for the following assays.

#### 3.7.2. Effect of PP on NO Production

NO is a lipid-soluble gaseous molecule with free radical properties, serving as a crucial inflammatory mediator involved in various biochemical reactions. It has been recognized as a key indicator of macrophage activation [51]. As demonstrated in Figure 3B, LPS stimulation (1 μg/mL) considerably raised NO production in RAW264.7 macrophages to 5.42 times that of the control group, confirming the effective creation of a model of cellular inflammation. Compared to the LPS model group, 20 μg/mL PP pretreatment showed no significant inhibitory impact on LPS-triggered upregulation of NO (*p* > 0.05). However, 100 μg/mL and 500 μg/mL PP pretreatments remarkably inhibited the LPS-induced release of NO by 40.62% and 68.61%, respectively, demonstrating a dose-dependent trend within the tested range. Xu et al. [52] found that chlorogenic acid exhibited NO-inhibitory activity in LPS-triggered RAW264.7 cells, with significant suppression observed at 12.5 μg/mL. In our study, while the concentration of chlorogenic acids in 100 μg/mL PP exceeded the threshold, the observed effect may not be solely attributed to chlorogenic acids, as other polyphenolic components may also contribute to the inhibition of NO production. This is supported by the findings of Lee et al. [53], who demonstrated that both quercetin and rutin significantly reduced NO levels at 100 μg/mL (*p* < 0.05) in LPS-activated RAW264.7 macrophages. Moreover, the structural differences of SPSL polyphenols (phenolic acids and flavonoids) may enable them to synergistically enhance anti-inflammatory effects by targeting multiple pathways, such as inhibiting NF-κB and modulating PI3K/Akt (e.g., quercetin) [54], thereby reducing NO levels.

#### 3.7.3. Effect of PP on Inflammatory Cytokine Expression and Release

Activated macrophages secrete various inflammatory cytokines, thereby regulating cellular and humoral immune responses [55]. To explore the regulatory impacts of PP on LPS-induced inflammatory cytokines’ expression and release in RAW264.7 macrophages, mRNA abundance and cytokine levels were analyzed via RT-qPCR and ELISA, respectively. As illustrated in Figure 4, LPS exhibited dual regulation. It induced a robust upregulation of IL-1β, IL-6, and TNF-α at both the protein (*p* < 0.0001) and transcriptional levels (*p* < 0.001), while simultaneously suppressing IL-10 production and mRNA expression (*p* < 0.0001) compared to the controls. These outcomes indicate that LPS effectively triggered macrophage inflammatory activation, resulting in a dominance of pro-inflammatory cytokines. In comparison with the LPS model group, both the LPS + PP (100 μg/mL) and LPS + PP (500 μg/mL) groups exhibited significant reductions in the release and transcript levels of IL-1β, IL-6, and TNF-α (*p* < 0.01 for 100 μg/mL; *p* < 0.001 for 500 μg/mL). Likewise, Du et al. [56] reported that pomegranate peel polyphenols suppressed pro-inflammatory cytokine gene expression and release in LPS-induced RAW264.7 cells. Additionally, PP pretreatment (100 and 500 μg/mL) significantly promoted IL-10 release while elevating its gene expression levels (*p* < 0.0001). Comalada et al. [57] demonstrated that certain flavonoid compounds (e.g., quercetin, luteolin) can enhance IL-10 release while suppressing TNF-α and IL-1β. The purified SPSL polyphenols in our study contain this type of polyphenolic compound, which is likely the reason for the observed results. These findings demonstrate that PP exhibits anti-inflammatory effects by suppressing pro-inflammatory cytokine expression and release while promoting anti-inflammatory cytokine production.

#### 3.7.4. Effect of PP on NF-κB Pathway Stimulation

The NF-κB signaling pathway is an important intracellular signaling cascade that regulates inflammatory responses. Under resting conditions, IκBα forms a non-phosphorylated complex with NF-κB p65, and they co-localize in the cytoplasm. Upon stimulation by upstream signals, the IκB kinase (IKK) complex gets activated, resulting in the IκBα phosphorylation at Ser32/36. The phosphorylated IκBα subsequently undergoes ubiquitination and proteasomal degradation, which illustrates the nuclear localization signal of the NF-κB p65 subunit. This triggers conformational changes in NF-κB p65, facilitating its translocation into the nucleus to stimulate target genes’ transcription [39]. Based on this molecular mechanism, proteins in the cytoplasm and nucleus were isolated from RAW264.7 cells to assess IκBα phosphorylation/degradation and NF-κB p65 nuclear translocation.

Figure 5A shows that LPS induced a 5.95-fold enhancement (*p* < 0.001) in NF-κB p65 nuclear localization compared to controls, confirming effective nuclear translocation. However, PP pretreatment dose-dependently reduced the progression of this process, with 100 μg/mL and 500 μg/mL PP reducing translocation by 45.42% and 86.67%, respectively. The NF-κB p65 nuclear accumulation depends on the phosphorylation and subsequent IκBα degradation. Compared with the control group, a 7.36-fold induction of IκBα phosphorylation (*p* < 0.0001) and 84.10% degradation of IκBα (*p* < 0.001) were observed following LPS treatment, as visualized in Figure 5B,C. In contrast, PP pretreatment dose-dependently suppressed LPS-stimulated IκBα phosphorylation, with 20, 100, and 500 μg/mL PP reducing phosphorylation by 53.40%, 67.55%, and 83.63%, respectively. It also inhibited IκBα degradation by 23.16% (100 μg/mL) and 53.91% (500 μg/mL).

These outcomes indicate that PP may display anti-inflammatory action by inhibiting the IκBα phosphorylation and degradation, thereby suppressing NF-κB signal transduction. This is concordant with the discoveries of Zhang et al. [34], who reported that 0.2 mg/mL purified sweet potato leaf phenolic acid extract significantly reduced IL-1β-induced nuclear translocation of p65 in Caco-2 cells and increased the stability of IκBα, indicating an anti-inflammatory impact by the NF-κB pathway suppression. Furthermore, Francisco et al. [58] confirmed that chlorogenic acid, the key phenolic acid component in lemongrass (*Cymbopogon citratus*) leaf extract, targeted the inhibition of IκBα phosphorylation and proteasome-mediated degradation in an LPS-induced macrophage model, thereby blocking p65 nuclear translocation. This inhibition downregulated NF-κB-driven gene transcription, ultimately causing a substantial decline in pro-inflammatory cytokine mRNA expression. In this study, we determined NF-κB p65 nuclear translocation by analyzing its protein levels in the nucleus. Meanwhile, related studies have employed immunofluorescence to directly visualize the subcellular localization of p65 [59]. A combination of these two methods would provide more robust and complementary evidence. The nuclear translocation of NF-κB p65 was inhibited by PP pretreatment, leading to reduced NF-κB p65 phosphorylation and downstream pro-inflammatory gene expression, ultimately decreasing the release of pro-inflammatory mediators (NO, IL-1β, IL-6, and TNF-α).

#### 3.7.5. Influences of PP on MAPK Pathway Activation

Classic MAPK members comprise extracellular regulated protein kinase (ERK), p38, and c-Jun NH2-terminal kinase (JNK) [60,61]. In LPS-stimulated macrophages, MAPK signaling critically regulates inflammatory mediator production and release [62]. To evaluate MAPK pathway stimulation, the key signaling proteins’ expression was assessed via Western blot. Figure 6A–C illustrates that LPS stimulation significantly upregulated the phosphorylation levels of ERK, p38, and JNK, indicating the triggering of the MAPK cascade. PP pretreatment dose-dependently inhibited the phosphorylation of these three kinases. In comparison with the LPS model group, the p-ERK/ERK, p-p38/p38, and p-JNK/JNK ratios in the LPS + 100 μg/mL PP group decreased by 44.68%, 57.87%, and 42.34%, respectively (*p* < 0.001), while those in the LPS + 500 μg/mL PP group showed more pronounced reductions of 67.97%, 72.03%, and 58.29%, respectively (*p* < 0.0001). These outcomes suggest that PP exerts its anti-inflammatory action by potently blocking ERK, JNK, and p38 activation in the MAPK pathway. Guo et al. [31] found that bound polyphenols from the insoluble dietary fiber of navel orange peel suppressed the LPS-induced phosphorylation of ERK, JNK, and P38 in a dose-dependent manner, indicating that its anti-inflammatory properties may be mediated through the MAPK pathway, which was similar to our research findings. PP-mediated inhibition of MAPKs (p38, ERK, JNK) may attenuate activator protein (AP)-1 and activating transcription factor 2 (ATF2) activation while enhancing tristetraprolin (TTP)-dependent mRNA degradation, thereby reducing the production of pro-inflammatory cytokines [63,64].

## 4. Conclusions

This study optimized the purification process using NKA-II macroporous resin and examined the anti-inflammatory activity of the purified polyphenols via an LPS-triggered RAW264.7 macrophage model. Purification with NKA-II macroporous resin achieved a polyphenol recovery rate of 85.30 ± 0.85% and a purity of 75.70 ± 0.92% in the final product. Pilot-scale experiments confirmed that the NKA-II resin purification process is feasible for industrial applications. The LPS-stimulated RAW264.7 model findings provide strong verification for the anti-inflammatory activity of SPSL polyphenol. However, this model cannot fully replicate in vivo macrophage polarization and lacks physiological barriers (e.g., the blood–brain barrier), which may alter polyphenol pharmacokinetics. Therefore, further in vivo studies are essential to validate its anti-inflammatory mechanisms and comprehensively evaluate efficacy, safety, and potential limitations such as metabolic degradation and bioavailability. Notably, this study proposes a novel framework for the high-value utilization of SPSL in the food industry. SPSL polyphenols purified using NKA-II resin could be nanoencapsulated and incorporated into various food matrices, such as beverages and yogurt, to develop functional foods aimed at mitigating chronic inflammation.

## Figures and Tables

**Figure 1 foods-14-02903-f001:**
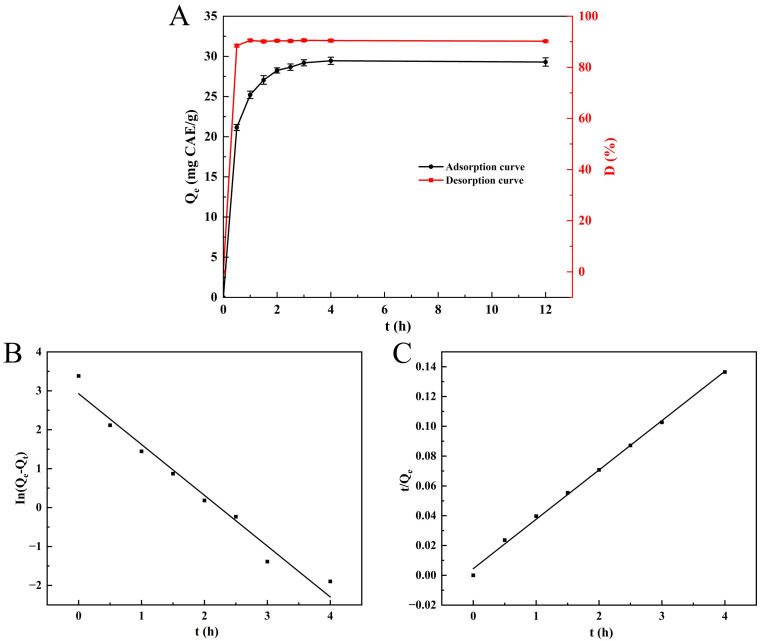
Static adsorption/desorption curve of SPSL polyphenols on NKA-II resin (**A**); pseudo-first-order kinetic model (**B**); and pseudo-second-order kinetic model (**C**). *Q*_e_ was the equilibrium adsorption capacity; *Q*_t_ was the adsorption capacity at time t; *D* was the desorption ratio.

**Figure 2 foods-14-02903-f002:**
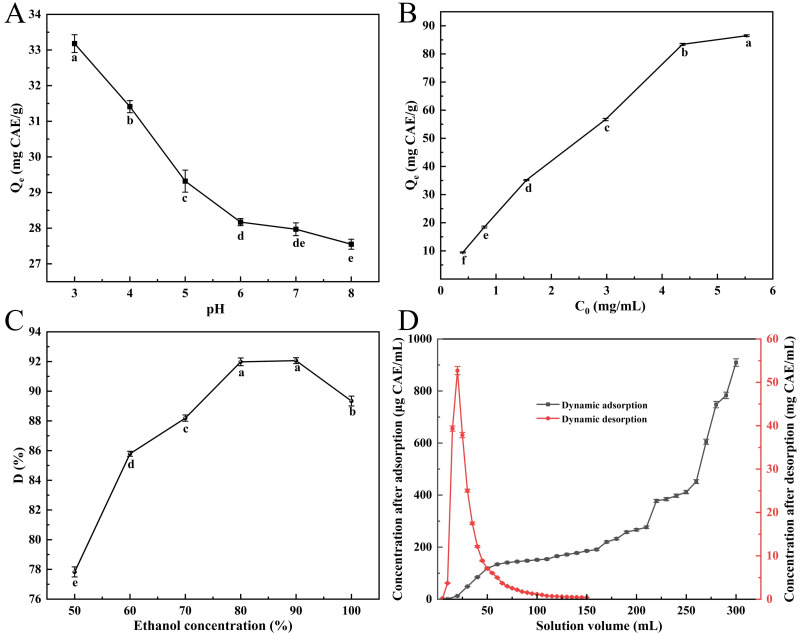
Influence of pH on NKA-II resin adsorption of polyphenols (**A**); effect of initial concentration on SPSL polyphenol uptake by NKA-II (**B**); effect of eluent concentration on desorption ratio of SPSL polyphenols on NKA-II resin (**C**); and dynamic adsorption/desorption curves of SPSL polyphenols on NKA-II resin (**D**). *C_0_* was the initial total phenolic concentration of sample solution; *Q_e_* and *D* were the same as described in Figure 1. Differing lowercase letters in (**A**–**C**) indicate statistically significant differences (*p* < 0.05).

**Figure 3 foods-14-02903-f003:**
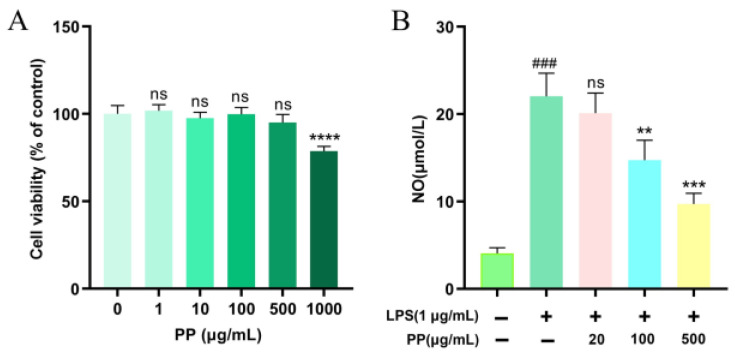
Effect of PP on the viability and NO production of RAW264.7 cells. (**A**): effect of PP on the viability of RAW264.7 cells, ^ns^
*p* > 0.05, **** *p* < 0.0001 vs. PP (0 μg/mL); (**B**): effect of PP on the NO production of RAW264.7 cells. RAW264.7 cells were pretreated with PP (0, 20, 100, and 500 μg/mL, respectively) for 1 h before they were incubated with LPS (1 μg/mL) for 24 h. ### *p* < 0.001 vs. control group (0 μg/mL LPS and 0 μg/mL PP); ^ns^ *p* > 0.05, ** *p* < 0.01, and *** *p* < 0.001 compared to the LPS-stimulated group.

**Figure 4 foods-14-02903-f004:**
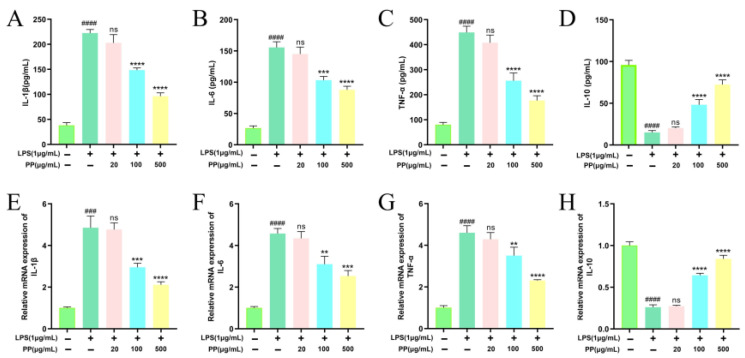
Effects of PP on LPS-induced release of IL-1β (**A**), IL-6 (**B**), TNF-α (**C**), and IL-10 (**D**); effects of PP on LPS-induced gene expression of IL-1β (**E**), IL-6 (**F**), TNF-α (**G**), and IL-10 (**H**). Different colors represent different treatments, with the corresponding treatments shown on the horizontal axis. Differences as compared with the control group (### *p* < 0.001, #### *p* < 0.0001) and LPS model group (^ns^ *p* > 0.05, ** *p* < 0.01, *** *p* < 0.001, **** *p* < 0.0001).

**Figure 5 foods-14-02903-f005:**
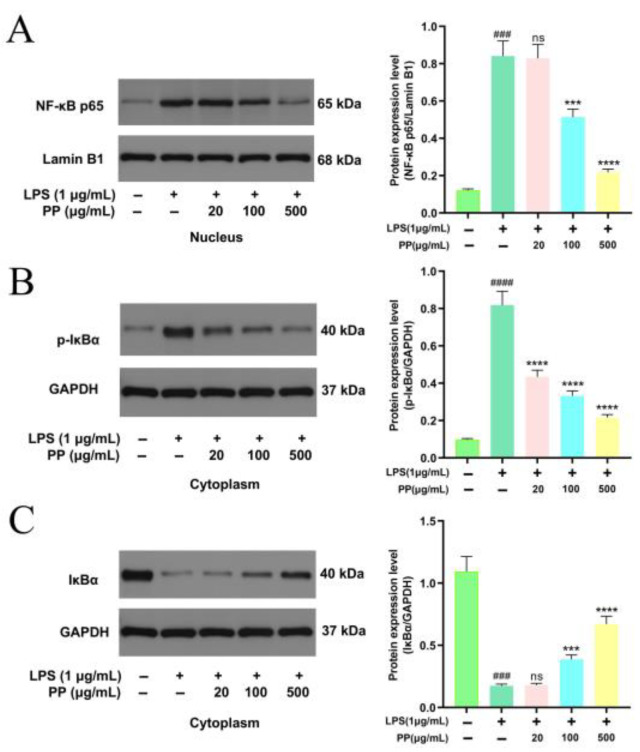
Representative bands and relative intensities of the NF-κB signaling pathway. (**A**): NF-κB p65 in the nucleus; (**B**): *p*-IκBα in the cytoplasm; and (**C**): IκBα in the cytoplasm. In terms of relative intensities, different colors represent different treatments, as indicated on the horizontal axis. Differences as compared with the control group (### *p* < 0.001, #### *p* < 0.0001) and LPS model group (^ns^ *p* > 0.05, *** *p* < 0.001, **** *p* < 0.0001).

**Figure 6 foods-14-02903-f006:**
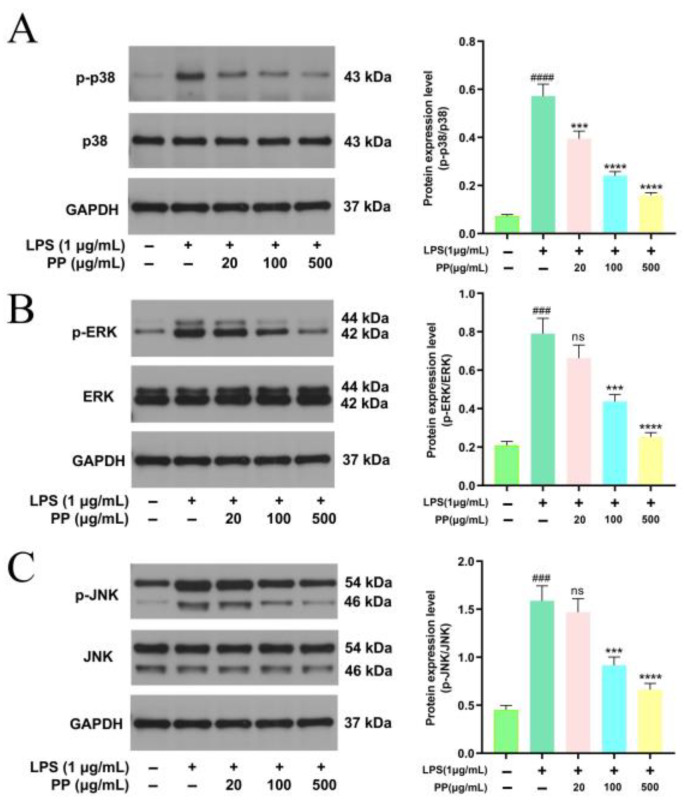
Representative bands and relative intensities of the MAPK signaling pathway. (**A**): p-p38/p38; (**B**): p-ERK/ERK; and (**C**): p-JNK/JNK. In terms of relative intensities, different colors represent different treatments, as indicated on the horizontal axis. Differences as compared with the control group (### *p* < 0.001, #### *p* < 0.0001) and LPS model group (^ns^ *p* > 0.05, *** *p* < 0.001, **** *p* < 0.0001).

**Table 1 foods-14-02903-t001:** Physical and adsorption/desorption properties of the tested macroporous resins.

Resin Types	Polarity	Specific Surface Area (m^2^/g)	Average Pore Size (nm)	Adsorption Capacity (mg CAE/g)	Desorption Rate (%)
S-8	Polar	100–120	28–30	22.85 ± 0.11 ^b^	88.48 ± 0.99 ^b^
NKA-9	Polar	170–250	15.5–16.5	21.17 ± 0.41 ^c^	83.83 ± 0.35 ^d^
NKA-Ⅱ	Polar	950–1250	14.5–15.5	29.32 ± 0.19 ^a^	90.50 ± 0.28 ^a^
DA-201	Polar	≥200	10–13	21.15 ± 0.52 ^c^	86.26 ± 0.89 ^c^
AB-8	Weak polar	480–520	13–14	22.45 ± 0.33 ^b^	86.12 ± 0.53 ^c^
HP20	Non-polar	≥500	75–80	21.34 ± 0.01 ^c^	89.39 ± 0.34 ^ab^

Note: the presence of differing superscript letters in columns 5–6 reflects significant differences (*p* < 0.05); the physical properties of the six tested resins (columns 2–4) were obtained from the manufacturer’s specifications.

**Table 2 foods-14-02903-t002:** Kinetic analysis of SPSL polyphenol adsorption onto NKA-II resin.

Models	Equations	Parameters		
PFO *	In (Q_e_−Q_t_) = −1.30556 t + 2.92608	R^2^ = 0.9671	k_1_ = 1.3056 h^−1^	Q_e_ = 18.654 mg CAE/g
PSO **	t/Q_t_ = 0.03314 t + 0.00442	R^2^ = 0.9971	k_2_ = 0.2485 g·mg^−1^·h^−1^	Q_e_ = 30.175 mg CAE/g

Note: * refers to pseudo-first-order, ** refers to pseudo-second-order.

**Table 3 foods-14-02903-t003:** Characterization of purified SPSL polyphenols by UHPLC-QE-MS/MS.

No.	Rt (min)	[M-H]^−^ *(m*/*z)*	MS Fragments (*m*/*z*)	Molecular Formula	Identification
Phenolic acids					
2	4.44	353.0871	191.0554, 179.0341	C_16_H_18_O_9_	5-Caffeoylquinic acid
3	4.87	339.0719	177.0186, 161.0235	C_15_H_16_O_9_	Esculin
5	5.49	179.0341	135.0440	C_9_H_8_O_4_	Caffeic acid
6	5.69	161.0235	133.0284	C_9_H_6_O_3_	7-Hydroxycoumarin
7	5.89	353.0871	191.0555, 173.0447	C_16_H_18_O_9_	3-Caffeoylquinic acid
8	6.74	353.0871	191.0553	C_16_H_18_O_9_	4-Caffeoylquinic acid
9	7.51	367.1025	191.0553, 134.0358,	C_17_H_20_O_9_	Feruloylquinic acid
12	9.01	207.0657	163.0757	C_11_H_12_O_4_	Ethyl caffeate
13	9.18	515.1179	353.0878, 173.0446	C_25_H_24_O_12_	3,4-Dicaffeoylquinic acid
14	9.52	515.1179	353.0878, 191.0553	C_25_H_24_O_12_	3,5-Dicaffeoylquinic acid
15	10.04	515.1179	353.0875, 179.0340	C_25_H_24_O_12_	4,5-Dicaffeoylquinic acid
17	12.14	677.1487	515.1185, 173.0446	C_34_H_30_O_15_	3,4,5-Tricaffeoylquinic acid
Flavonoids					
1	4.01	301.0351	257.4846	C_15_H_10_O_7_	Quercetin
4	5.00	285.0437	152.0106, 108.0205	C_15_H_10_O_6_	Kaempferol
10	8.11	609.1448	300.0287	C_27_H_30_O_16_	Rutin
11	8.61	463.0872	300.0276, 151.0028	C_21_H_20_O_12_	Hyperoside
16	12.13	299.0555	284.0320, 151.0023	C_16_H_12_O_6_	Diosmetin
18	12.38	329.0662	314.0436, 299.0203	C_17_H_14_O_7_	Jaceosidin
19	14.36	313.0714	298.0487	C_17_H_14_O_6_	Pectolinarigenin

**Table 4 foods-14-02903-t004:** Content of chlorogenic acids in the purified SPSL polyphenols.

Peak No.	Retention Time (min)	Identification	Standard Curve	R^2^	Content (mg/g DW)
1	6.218	5-CQA	y = 24,700 x + 979	0.999977	17.39 ± 0.24
2	10.547	3-CQA	y = 30,000 x – 32,900	0.999945	113.35 ± 1.26
3	11.078	4-CQA	y = 30,800 x − 8190	0.999987	17.14 ± 0.34
4	15.972	3,4-diCQA	y = 30,900 x – 40,300	0.999975	75.52 ± 0.53
5	16.563	3,5-diCQA	y = 37,300 x – 56,900	0.999968	54.21 ± 0.28
6	16.951	4,5-diCQA	y = 34,500 x − 6460	0.999984	11.37 ± 0.13
7	19.800	3,4,5-triCQA	y = 30,000 x – 92,700	0.999208	9.03 ± 0.10
Sum					298.01 ± 2.88

## Data Availability

The original contributions presented in the study are included in the article and Appendix A, further inquiries can be directed to the corresponding authors.

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
