# Peer review of "Purification, Composition, and Anti-Inflammatory Activity of Polyphenols from Sweet Potato Stems and Leaves"

_foods, 2025, doi:10.3390/foods14162903_

Round 1
Reviewer 1 Report
Comments and Suggestions for Authors
The manuscript provides valuable insights into the extraction and bioactivity of polyphenols from sweet potato stems and leaves (SPSL). It is experimentally solid, and the purification and characterization techniques are well-chosen. However, from an organic chemistry perspective, there are areas where mechanistic insight, structural specificity, and analytical validation could be improved or clarified.
The main limitation in the Introduction is the lack of recent references (2023–2025) on the valorization of sweet potato by-products, including stems, leaves, and especially peels, which are increasingly recognized for their polyphenolic content and functional potential. Incorporating up-to-date literature would significantly enhance the timeliness and relevance of the study. Additionally, it would be valuable to highlight that sweet potato peels are considered part of the plant’s agro-industrial waste stream and are utilized by some researchers in polyphenol recovery studies.
Here are my other suggestions:
Lines 12–27: The abstract could be improved by quantifying the extent of anti-inflammatory activity. For example, compare the efficacy to a known control or standard anti-inflammatory agent to contextualize the findings.
Lines 45–46: The claim regarding limited pharmacological evaluations of SPSL polyphenols would be stronger if supported by citations or relevant data indicating gaps in the literature.
Lines 64–65: Consider elaborating on how molecular properties (e.g., hydrophilicity, aromaticity, or hydroxyl group distribution) influence adsorption efficiency on different resins. This would enhance the rationale for selecting NKA-II.
Lines 163–174: Mass spectrometry protocols are thorough, but the manuscript should specify the criteria used to identify unknown compounds (e.g., fragmentation patterns, exact mass tolerances) to improve reproducibility and chemical clarity.
Lines 401–416: The HPLC method used for chlorogenic acid quantification lacks essential validation metrics such as recovery rate, linearity, precision, and LOD/LOQ. These are critical for evaluating the reliability of quantification.
Line 449: The anti-inflammatory activity is attributed to chlorogenic acids (CGAs), but it remains unclear whether these effects stem solely from CGAs or from synergistic interactions with other flavonoids. Clarification here would be useful.
Lines 503–504: The discussion of NF-κB inhibition would benefit from deeper mechanistic detail. For instance, was nuclear translocation of p65 directly visualized (e.g., via immunofluorescence), or inferred from downstream markers alone?
Lines 542–554: The conclusion would be stronger if it acknowledged limitations such as the lack of in vivo validation, potential metabolic degradation of polyphenols, or bioavailability concerns.
Reviewer 2 Report
Comments and Suggestions for Authors
The study titled “Purification, Composition, and Anti-Inflammatory Activity of Polyphenols from Sweet Potato Stems and Leaves” investigates the enrichment of polyphenols from sweet potato stems and leaves (SPSL) using macroporous resins, their characterization via UHPLC-QE-MS/MS, and the evaluation of their anti-inflammatory activities. The results demonstrate that NKA-II resin achieved the highest purification efficiency, chlorogenic acids were the predominant compounds, and PP exhibited notable anti-inflammatory effects by inhibiting MAPK and NF-κB pathways. The study is considered novel, and it is recommended that the manuscript be improved with the suggested revisions below.
-A brief discussion could be added regarding a prominent factor, among those potentially correlating with the adsorption degree. Line 250.
-In Fig. 1a, is there a lack of data for the initial hours in the desorption curve?
-For statistical analyses, one-way ANOVA followed by appropriate multiple comparison tests (e.g., Tukey’s post-hoc) should be applied to determine significant differences among groups.
-In the manuscript overall, while the optimization of the purification process is clearly stated, the criteria and statistical approach used to define the “optimum” conditions (e.g., response surface methodology, ANOVA) are not specified. These should be clarified to ensure reproducibility.
Reviewer 3 Report
Comments and Suggestions for Authors
1. A specific projection or application should be added at the end of the summary to strengthen its impact and practical relevance.
2. Improve the keywords by including more specific terms to improve the thematic accuracy and visibility of the article.
3. In the introduction, improve the wording regarding the novelty of the study and more clearly define the scientific gap related to the anti-inflammatory mechanisms of SPSL polyphenols.
4. Ensure that all abbreviations are fully described the first time they appear in the manuscript.
5. The materials and methods section should include information on all materials, supplies, and equipment used in the research, including, where applicable, the model, brand, country, and city of manufacture.
6. Section 2.2 lacks a methodological source.
7. Information on replicability in dynamic purification is missing, positive and negative controls are not indicated in the cell assays, and details of the validation of the analytical method by HPLC are not reported.
8. Section 2.3 should be explicitly detailed, including the equation used.
9. Sections 2.4.3, 2.4.4, and 2.6 lack methodological sources. Check the others that do not have a methodological source and provide as much detail as possible for replication.
10. In section 2.8, explicitly indicate the experimental design used and the specific techniques used for that design. In addition, include whether all statistical assumptions have been tested.
11. Table 1 should include explanatory footnotes.
12. For Figure 1, no other graphic representation software, such as Origin or similar, was used. If this was the case, include in section 2.8 which software was used.
13. Explain in greater depth the reason for the low performance of polarized resins such as NKA-9 compared to NKA-II.
14. Include a comparative statistical analysis between the types of resins evaluated to strengthen the selection.
15. Discuss why the pseudo-second-order kinetic model is more appropriate from a molecular point of view.
16. Better justify the choice of pH 3.0, considering the stability of polyphenols in acidic media.
17. Compare the chlorogenic acid content with other similar studies to contextualize its abundance.
18. Indicate whether there was a correlation between polyphenol content and the anti-inflammatory activity observed.
19. Expand the discussion on the specific role of individual compounds, such as quercetin or rutin, in the effects observed.
20. Include a more critical analysis of the limitations of the RAW264.7 model for extrapolating results to in vivo conditions.
21. Incorporate a discussion on the possible synergy between the polyphenols present in the purified mixture.
22. Reinforce the industrial projection of NKA-II resin based on costs, scalability, and reuse.
23. In addition to making comparisons with other studies, the biological, physical, and chemical mechanisms involved in all the results obtained should be explored.
24. Discuss whether the process is viable at an industrial level, considering aspects such as costs, scalability, and impact on product quality.
25. It would be advisable to include more recent scientific articles (preferably from the last five years) that support and enrich the discussion of the results.
26. The conclusions should be improved by including the limitations of the study and possible new lines of research. The potential for application in the food industry should also be highlighted.
27. It is recommended to reduce the similarity index of the iThenticate software (22%), especially in the methodology, results, and discussion sections.
Round 2
Reviewer 3 Report
Comments and Suggestions for Authors
Accepted in the present form.